# Targeting the Multiple Complex Processes of Hypoxia-Ischemia to Achieve Neuroprotection

**DOI:** 10.3390/ijms25105449

**Published:** 2024-05-17

**Authors:** Auriane Maïza, Rifat Hamoudi, Aloïse Mabondzo

**Affiliations:** 1CEA, DMTS, SPI, Neurovascular Unit Research & Therapeutic Innovation Laboratory, Paris-Saclay University, CEDEX 91191 Gif-sur-Yvette, France; auriane.letrou@cea.fr; 2Center of Excellence of Precision Medicine, Research Institute for Medical and Health Sciences, University of Sharjah, Sharjah P.O. Box 27272, United Arab Emirates; rhamoudi@sharjah.ac.ae; 3College of Medicine, University of Sharjah, Sharjah P.O. Box 27272, United Arab Emirates; 4Division of Surgery and Interventional Science, University College London, London NW3 2PF, UK

**Keywords:** hypoxia-ischemia, blood-brain barrier, inflammation, neuroprotection, therapy

## Abstract

Hypoxic-ischemic encephalopathy (HIE) is a major cause of newborn brain damage stemming from a lack of oxygenated blood flow in the neonatal period. Twenty-five to fifty percent of asphyxiated infants who develop HIE die in the neonatal period, and about sixty percent of survivors develop long-term neurological disabilities. From the first minutes to months after the injury, a cascade of events occurs, leading to blood-brain barrier (BBB) opening, neuronal death and inflammation. To date, the only approach proposed in some cases is therapeutic hypothermia (TH). Unfortunately, TH is only partially protective and is not applicable to all neonates. This review synthesizes current knowledge on the basic molecular mechanisms of brain damage in hypoxia-ischemia (HI) and on the different therapeutic strategies in HI that have been used and explores a major limitation of unsuccessful therapeutic approaches.

## 1. Introduction

Hypoxia-ischemia (HI) is caused as a result of insufficient blood flow to the brain combined with lower-than-normal concentrations of oxygen in the arterial blood during the neonatal period [1]. Different factors can cause HI and the origin can be maternal (medical failure, medication errors, infections, labor and delivery errors), fetal (stroke, fetal growth restriction, infection, neonatal health mismanagement) or placental/uterine (placenta previa, placental abruption, placental insufficiency, oligohydramnios or polyhydramnios, umbilical cord prolapse, uterine rupture or nuchal cord) [2,3]. Two and a half per 1000 infants experience HI at birth resulting in hypoxic-ischemic encephalopathy (HIE) and neurodevelopmental disabilities that place a lifelong burden on parents and society [4]. This event in the immature brain can cause significant morbidity, mortality, and result in long-term neurological deficits including cerebral palsy, epilepsy, intellectual deficits, severe learning difficulties, speech and motor disorders and behavioral disabilities [5,6,7,8,9,10].

Neonatal HIE is the most common cause of death and disability in neonates, accounting for 23% of infant mortality worldwide [11]. At birth, the Sarnat grading scale is used to classify newborns suffering from HIE in three different classes from mild to severe forms of the disease by an evaluation of the symptoms shown in Table 1 [12]. Most of the time, the diagnosis of HIE is not proven but is assumed from a variety of markers that reflect hypoxia and ischemia. In the mild form of the disease, the prognosis is favorable and spontaneous recovery is viable for all patients. Neonates with mild encephalopathy usually do not have an increased risk of motor or cognitive deficits. In the moderate form, 75% of neonates spontaneously recover physiologic functions, whereas 25% develop long-term disabilities. Neonates with moderate encephalopathy have significant memory impairment, visual or motor dysfunction and increased hyperactivity [13]. In severe HIE, the mortality rate is between 25 and 50% [11,14] and most deaths occur in the first days after birth. Neonates with severe HIE have an increased risk of intellectual disability [7]. The incidence of long-term neurological disabilities depends on the severity of HIE. As many as 60% of infants who survive moderate or severe HI develop serious long-term complications such as mental disorders, seizures, learning disabilities, cerebral palsy (10–20%), speech disorder, visual and auditory problems (about 40%) (Figure 1).

Despite a large number of clinical trials, no pharmacological treatment is available to attenuate brain injury in neonates. Therapeutic hypothermia (TH) is the only approved therapy to treat HIE, but it can only be used in full-term infants. Moreover, TH is only partially protective and mainly attenuates injury only after moderate HIE [15,16]. It inhibits many steps of the oxidative cascade and protects the brain from the spread of injury. Initiated within 6 h of birth, the brain temperature is lowered to 33–34 °C and maintained for 72 h. Recent reports suggest that current TH protocols are near-optimal [17]. Unfortunately, TH is applicable only in term infants (at least 36 weeks gestational age) in moderate cases, or in some cases severe disease with specific inclusion criteria (time lapse after the injury <6 h, umbilical cord pH < 7, etc.) [18]. Moreover, TH requires specific and expensive equipment and investment which can be difficult for hospitals in low-income countries [19].

In support of the findings in infants with HIE treated with TH, studies in rodents also suggest that hypothermia is not neuroprotective after exposure to severe HI [20]. It reduces the risk of death and severe disability after neonatal HI by only 12 to 15% at 18 months after the insult and is ineffective for a large number of neonates [20].

Therapies other than supportive care are not available to treat brain injury in preterm infants exposed to HI.

Consequently, there is an urgent need for novel therapeutic options to treat premature and full-term infants with HIE. In this review, we highlight the early response to brain injury in HI and, while focusing on the main target of HI, we describe the past and current therapeutic interventions in HI. Finally, we point to future directions in drug development for the treatment or attenuation of hypoxic-ischemic brain injury.

## 2. Early Response to Brain Injury in Hypoxic-Ischemia

Following the HI event, the evolution of brain injury is a dynamic process that may continue for days to weeks, months and years after the insult. Within the first few hours, in the acute phase of the disease, decreased cerebral blood flow reduces delivery of oxygen and glucose to the brain, which are normally used in oxidative phosphorylation to produce adenosine triphosphate (ATP) [21]. Because of this depletion in ATP, the Na^+^/K^+^ pump fails and lactate production is increased to provide energy but also to help brain recovery [22,23]. The pump failure triggers depolarization of neurons leading to release of glutamate which accumulates in the inter-synaptic space [19,24]. Glutamate is an excitatory mediator that binds to postsynaptic receptors such as the N-methyl-D-aspartate (NMDA), α-amino-3-hydroxy-5-methyl-4-isoxazolepropionic acid (AMPA) and kainate receptors triggering their over-activation leading to massive calcium entry into the post-synaptic element [24,25].

The early response of the brain to HI is characterized by necrotic cell death resulting from excitatory neurotoxicity, production of oxygen radicals, oxidative stress, and mitochondrial dysfunction [26,27,28]. Post-ischemic inflammation is a critical component and strong predictor in the development of brain injury after HI [27,29,30]. The inflammatory reaction appears a few hours after the insult and includes systemic and local release of several pro-inflammatory mediators called cytokines, including tumor necrosis factor (TNF-α), interleukin 1β (IL-1β) and interleukin 6 (IL-6). The cytokines are produced locally by microglia or infiltrated cells after crossing the blood-brain barrier (BBB) [28]. The increased activation of microglia and angiogenesis are early events in the pathology that partly regenerate brain tissue [31]. However, prolonged inflammation may affect brain development with long-lasting consequences resulting in neurologic disorders [32,33]. Thus, inflammation is recognized as a key factor in the pathophysiology of hypoxia-ischemia brain injury.

## 3. Blood-Brain Barrier and HI Brain Injury

The BBB is the cellular structure that separates the brain parenchyma from the bloodstream. This important barrier is predominantly composed of a specialized monolayer of brain endothelial cells lining the brain capillaries that are joined by well-developed tight junctions (TJs), pericytes, and astrocyte “end feet”. The BBB function requires the coordination between the different cell type mentioned above but also interaction with the extracellular matrix (ECM) in close contact with the BBB [34]. Specific features of the endothelium enable it to regulate the exchange between the blood and brain. Thus, the BBB constantly regulates the extracellular environment of the brain which is essential for the smooth functioning of synaptic transmission between neurons but also constitutes a physical and functional barrier to prevent some toxic chemicals or biologics from crossing into the central nervous system (CNS). The physical barrier is a result of very limited paracellular passage explained by the presence of TJs between neighboring brain endothelial cells. These protein junctions connect to the cytoskeletal actin protein to form a continuous membrane with high electrical resistance [35]. TJs are complex protein networks composed of two major proteins: claudins and occludin, which interact with accessory proteins including zonula occludens [35] that are anchored to the actin cytoskeleton. Dynamic interactions between the actin cytoskeleton and junctional proteins are critical for the regulation of junctional integrity and endothelial permeability. The endothelium also contains a wide variety of transporters that regulate the passage of larger molecules. Early in development, well-formed TJs between brain endothelial cells have been shown to constitute the physical basis for effective barrier mechanism(s) and are crucial to establish a stable brain environment needed for proper brain maturation [36]. The brain endothelial cells also express efflux and influx transporters that regulate the brain environment [37]. Most alterations in the immature BBB are observed early (hours to days) after exposure to an HI insult [34,38,39]. A large number of in vitro and in vivo studies have documented increased BBB permeability with alterations in the expression and localization of key TJ components and transporters after HI in fetuses and neonates [39,40,41,42,43]. The consequences of increases in BBB permeability after HI are multi-factorial and involve features such as energy deprivation, release of reactive oxygen species (ROS) and local inflammation [34]. HI triggers the expression and release of proinflammatory mediators and immune cell chemo-attractants, and activates proteases. The microvascular inflammatory environment potentiates TJ disassembly in brain endothelial cells, increases the expression of adhesion molecules, and also damages the extracellular matrix, all of which results in BBB dysfunction [44]. The most widely reported effects of HI-related insults on the neurovascular unit (NVU) are increases in the permeability of the BBB. However, recent evidence also suggests the occurrence of significant alterations in metabolic function [34,39].

These facts suggest that HI-related inflammation and BBB/NVU abnormalities contribute to parenchymal brain injury. Both BBB dysfunction and inflammation represent important novel therapeutic targets in attempts to prevent brain injury after HI.

## 4. Therapeutic Strategies Currently Undergoing Clinical Trials

Several neuroprotective therapies are currently under development, such as melatonin, magnesium sulfate, and allopurinol. Unfortunately, few data are available concerning curative treatments, which is a reflection of the complexity of neuroregeneration. Table 2 and Table 3 summarize the current therapeutic approaches under development.

### 4.1. Allopurinol

Allopurinol is a xanthine oxidase inhibitor that has additional effects of scavenging toxic hydroxyl free radicals [45] and inhibiting neutrophil accumulation [46]. After administration, allopurinol is converted to oxypurinol, an even better hydroxyl radical scavenger, which crosses the BBB. Several preclinical studies in animal models suggest a neuroprotective effect of oxypurinol. High-dose allopurinol administered within a very short time (15 min) of recovery from cerebral HI markedly reduces both acute brain edema and long-term cerebral injury in immature rats [47]. Allopurinol has been reported to reduce hippocampal brain damage after acute birth asphyxia in late-gestation fetal sheep [48]. The ability of allopurinol to modulate BBB permeability during HIE is not described in the literature.

In 1998, van Bel et al. published the first study in neonates in which two doses of 20 mg/kg allopurinol or placebo were given after birth to 22 neonates. Neonates received the first dose up to 4 h after birth and a second dose 12 h later. The short-term effect of allopurinol was assessed with chemical biomarkers (lipid peroxidation and anti-oxidative parameters), the pattern of the cerebral blood flow was measured and electrical brain function was monitored with amplitude-integrated EEG. Uric acid levels were decreased from 16 h after birth in the treated group compared to the controls. However, lipid peroxidation and anti-oxidative parameters were the same in both groups. Moreover, allopurinol induced a relative preservation of cerebral blood flow and electrical brain activity was higher in the treated group, suggesting less brain damage. Unfortunately, the sample size of this study was small and the study was unblended. In later studies, the beneficial effect of allopurinol treatment was difficult to prove [49].

### 4.2. Magnesium Sulfate

Magnesium sulfate (MgSO_4_) blocks calcium channels. It is an inhibitor of glutamate release and a calcium channel antagonist involved in the maintenance of cell membrane permeability and mitochondrial functions [50]. MgSO_4_ is a non-competitive antagonist of the NMDA receptor. The neuroprotective effect of long-term MgSO_4_ administration after cerebral HI in newborn rats is related to the severity of brain damage [51].

In a clinical trial published in 2021, Iqbal et al. divided 62 neonates into two different groups [52]. The treated group (31 neonates) received three doses of MgSO_4_ (250 mg/kg) 24 h apart and the placebo group (31 neonates) received distilled water three times 24 h apart. The authors describe reductions in immediate disease complications, mortality and hospital stay. However, they did not detect any difference in neurodevelopmental outcome six months after HI.

### 4.3. Melatonin

Melatonin is a natural hormone involved in circadian rhythms and in different biological functions. It is primarily released by the pineal gland which regulates the sleep–wake cycle. Melatonin has a neuroprotective effect, through its ability to act as an antioxidant and anti-apoptotic by increasing the brain ATP level [53]. Melatonin has been reported to reduce inflammation and cell death in white matter in mid-gestation fetal sheep following umbilical cord occlusion [54]. Melatonin is also reported to have a beneficial effect on BBB permeability in hypoxic conditions [55].

In 2018, Ahmad et al. published data from a clinical trial with 80 newborns suffering from HI, treated with melatonin [56]. Newborns received 10 mg of melatonin orally within 12 h of birth and were followed up for 28 days to evaluate survival rates. In this clinical trial, melatonin was proposed as adjunctive therapy and led to an improved survival rate. In other studies, the beneficial effect of melatonin was not demonstrated and there is still controversy regarding the dose and the length of treatment needed to optimize the effect of melatonin on neonates suffering from HI.

### 4.4. Erythropoietin

Erythropoietin (EPO) is a molecule of high interest able to cross the BBB. EPO is a hormone secreted by the kidneys and the liver that increases the production rate of red blood cells in response to falling levels of oxygen in the tissues. In addition to multiple physiologic roles, EPO has multiple beneficial effects in various nervous system disorders through modulation of inflammatory cell activation [57,58] and stimulation of angiogenesis, and neurogenesis by increasing the secretion of VEGF and BDNF growth factors [59]. It has been shown in animal models that exogenous EPO exhibits neuroprotective effects by activation of anti-apoptotic, anti-oxidant and anti-inflammatory pathways and by stimulation of angiogenesis [60]. In cortical neuron cultures, it has been shown that EPO is able to reduce apoptosis induced by NMDA and NO [61] and limits the excitotoxic effect of glutamate and AMPA [62].

Since 2009 and the first clinical trial by Zhu et al., many clinical trials have been conducted to evaluate the efficacy of EPO in neonatal HI, following different treatment protocols [63,64]. In these different clinical trials, the doses ranged from 250 to 2500 U/kg with different modalities of administration (single or daily for 6 days), alone or in combination with TH. From these clinical trials we can conclude that treatment with EPO seems to be an effective strategy in combination or not in combination with TH, but many parameters need to be evaluated, such as dose, timing and frequency of administration. Moreover, this strategy seems to be ineffective for severe forms of the disease and shows gender-related differences in response to treatment [64]. A randomized controlled trial of high-dose EPO given intravenously for six doses followed by a maintenance doses three times per week up to 32 weeks of gestation for neuroprotection unfortunately did not result in lower risks of severe neurodevelopmental impairment or death at two years of age [65].

### 4.5. Dexmedetomidine

Dexmedetomidine is a α2-adrenergic receptor agonist that provides sedation, analgesia and neuroprotection through inhibition of capsase-3-mediated apoptosis induced by excitotoxicity [66]. Pretreatment with dexmedetomidine alleviated BBB disruption in an ischemic stroke model [67].

In 2021, Baserga et al. evaluated the neuroprotective effect of dexmedetomidine coupled with hypothermia in neonates after HI and compared the effect with that of morphine, which is another drug used to manage pain and sedation in this context [68]. They concluded that dexmedetomidine is a good alternative to morphine for the treatment of pain and sedation in newborns with HIE because of its neuroprotective properties [68].

### 4.6. Topiramate

Topiramate is an anticonvulsant able to block sodium channels and high voltage-activated calcium currents, as well as mitochondrial permeability transition pores, and also increases GABA-induced influx [69]. The efficacy of topiramate in reducing brain damage in HIE has been proven in animal models, though its ability to modulate BBB permeability after HI is not described in the literature.

Filippi et al. combined topiramate with hypothermia to evaluate its safety profile and efficacy in a clinical trial in term newborns with HIE. Newborns treated with 10 mg/kg topiramate once a day during the first three days of life showed a tendency for better seizure control and no adverse effects [70].

### 4.7. Xenon

Xenon is a noble and expensive anesthetic gas able to rapidly cross the BBB because of its very low blood-gas partition coefficient [71]. It decreases excitotoxicity by antagonism of NMDA, AMPA and kainate receptors [72]. It also participates in anti-apoptotic mechanisms, shows neuroprotective efficacy and is involved in important anti-inflammatory processes [73].

In 2010, Thoresen et al., recruited 12 infants with neonatal encephalopathy undergoing TH for a pilot study with xenon [74].

### 4.8. Citicoline

Citicoline is the exogenous form of cytidine-5-diphosphocholine. It has a neuroprotective effect probably linked to its ability to decrease glutamate-mediated cell death by the inhibition of neuronal glutamate efflux and the increase in astrocytic glutamate uptake [75,76]. In a traumatic brain injury model, citicoline reduced BBB breakdown [77]. This effect has not been proved in an HI model.

In a pilot project, Khushdil et al. intravenously (IV) administered citicoline at an unknown dose to twenty newborns with moderate to severe HIE [78]. The neonates were monitored during the immediate newborn period and the authors concluded that citicoline is a promising drug for the treatment of newborns with moderate to severe HIE [78]. The long-term effect of the molecule was not described in this study.

### 4.9. Autologous Umbilical Cord Blood Cells

Human umbilical cord blood (UCB) is an important reservoir of many different stem cell types, such as mesenchymal stem cells, hematopoietic stem cells and endothelial progenitor cells [79]. Stem cells transplantation in animal models induces significant functional recovery, and gives hope for neonatal HIE treatment. Several studies have shown regenerative effects in animal models [80]. In a review focusing exclusively on this approach, Serrenho et al. compiled data from 21 preclinical studies in which UCB cells were evaluated in HIE models [81]. Altogether, these studies revealed that treatment with UCB cells reduces apoptosis, neuronal loss, microglial activation, astrogliosis, and levels of pro-inflammatory cytokines, and promotes the expression of growth factors, angiogenesis, neuronal stem cell proliferation and neuron maturation [81]. Stem cell therapy is a promising HIE therapy. For example, Nabetani et al. showed that cell therapy may have a much longer therapeutic time window than TH because it might reduce apoptosis/oxidative stress and enhance the regenerative process [82].

Two major modes of action are involved in stem cell-mediated functional recovery in ischemic brain injury: cell replacement and the by-stander effect. Cellular and molecular neurorestorative mechanisms include neurogenesis, angiogenesis, synaptogenesis, immunomodulation, and trophic factor secretion [83,84].

While BBB abnormalities are an important novel therapeutic target in the prevention or attenuation of HI-related brain injury, no information is available on the effect of restoration of BBB function by the use of stem cells in HIE.

Table 2 and Table 3 summarize the therapeutic approaches that are currently undergoing clinical trials.
ijms-25-05449-t002_Table 2Table 2Current therapies and therapies under development: synthesis.
Current Therapies for NeuroprotectionNeuroprotective Therapies Undergoing Clinical TrailsTherapyHypothermiaMelatoninAllopurinolMagnesium sulfateErythropoietinTherapeutic effectsNeuroprotection[85]Neuroprotection [85]Neuroprotection[86]Neuroprotection[87]Neuroprotection[60]StatusCurrent therapyClinical TrialClinical TrialClinical TrialClinical TrialPatient profilesAcute perinatal asphyxia, infants are eligible >35 weeks of gestationNDNDTreatment recommended for premature infants rather than full-term newborn infantsNDSide effectsNo side effectsNDTeratogenic potentialSuspicion: mortality increaseNDMolecular mechanismsIncrease of cellular acetylation through acetyl-coA suppression [88]Inhibition of NO activityDecrease of NO concentration.Decrease of IL1β concentration and cytokines release.Decrease of caspase3 activation [85]Increase of ATP [53]Decrease of microglia activation [54]Elimination of free radicals and production of antioxidant enzymes [89,90]Xanthine oxydase inhibition [45]Hydroxyl radical reduction [47]Inhibition of neutrophils’ accumulation [46]Non-competitive NMDA receptor inhibitor [51,87]Decrease of cytokine production [54]Stimulation of oligodendrocytes differentiation [91]Erythropoietin reduces astrocyte activation and the recruitment of leukocytes and microglia [92]Increase of VEGF and BDNF growth factors [59]Blood-brain barrier/Able to cross the BBB[85]Able to cross the BBB[93]Able to cross the BBB[94]Able to cross the BBB[85]
ijms-25-05449-t003_Table 3Table 3Therapies under development: synthesis.
Under Neuroprotective Therapies Undergoing Clinical TrialsNeuroprotective and Regenerative TherapiesTherapyCombination of xenon and hypothermiaTopiramateN-acetylcysteineStem cellsCiticolineTherapeutic effectsNeuroprotection[73]Neuroprotection[95]Neuroprotection[96]NeuroprotectionRegenerative[97]NeuroprotectionRegenerative[75,98]StatusClinical TrialClinical TrialNDClinical TrialClinical TrialPatients’ profilesNDNDNDNDNDSide effectsSome side effects: Subcutaneous fat necrosis + transient desaturationNDNo side effects but low–level toxicityNDNDMolecular mechanismsNon-competitive NMDA receptor inhibitor [99]Stimulates anti-apoptotic factors [100]Kaïnates and AMPA receptors inhibitor [101]Calcium channels inhibitors [102], carbonic anhydrase isoenzyme [103] and «Mitochondrial permeability transition pore» protein inhibitor [104]Free radical elimination [105]Restores intracellular glutathion level [106]Decreases NO production [107]Stem cells promote neuronal growth in tissue regenerationDecrease inflammatory cells proliferation [108]Neuroprotective effect: inhibition of free fatty acid, stimulates phosphatidylcholine synthesis, diphosphatidyl glycerol and sphingomyelin preservation.Increase glutathione synthesis and glutathione reductase activity [85]Regenerative effect: unknown mechanism. Hypothesis: progenitor cell proliferation [85]Blood-brain barrierAble to cross the BBB[109]Able to cross the BBB[110]Able to cross the BBB[96]ND Able cross the BBB[111]


## 5. Neurotrophic Strategies under Development

During HI, the expression of many endogenous factors is modulated in the neonatal brain in order to limit brain damage. Of these, neurotrophins, an important family of trophic factors, are upregulated during HI and studied for decades as a therapeutic strategy in this context. Exogenous supplementation with some of them has been studied in vitro and or in vivo as potential approaches to the treatment of HIE.

### 5.1. Growth Hormone

Growth hormone (GH) is synthesized by somatotrophic cells in the anterior pituitary gland. As a neurotrophin, this hormone induces neuroprotection, neurite growth and synaptogenesis in response to neuronal injury. GH has been subcutaneously administered in Rice-Vannucci rat model at high doses (50 and 100 mg/kg) and provides moderate protection after HI (around 20%) [112]. In a mouse model of HI, three intraperitoneal injections of 4000 µg/kg (0 h, 12 h and 24 h after hypoxia) human-recombinant GH leads to modulation of inflammation and stabilization of the BBB by the modification of occludin expression [113].

### 5.2. Insulin-like Growth Factor-1

Insulin-like growth factor 1 (IGF-1) is a polypeptide hormone, a neurotrophic factor essential for the survival and functional maturation of immature neurons [114,115]. Evidence shows that endogenous IGF-1 is anti-apoptotic and promotes recovery after HI [115,116]. IGF-1 is also involved in cerebral angiogenesis during brain development [117]. In 2000, Pan et al. demonstrated that exogenous IGF-1 crosses the BBB and enters the CNS by a saturable transport system [118]. In a fetal sheep model of HI, 1 microgram of IGF-1 infused in the lateral cerebral ventricle over 1 h decreased neuronal loss [119]. However, because of its involvement in cerebral angiogenesis during development, the potential effect of this factor on BBB repair needs to be investigated in the context of HI.

### 5.3. Brain-Derived Neurotrophic Factor

Brain-derived neurotrophic factor (BDNF) is a neurotrophin, involved in the modulation of neuronal differentiation, growth and synaptic formation, activity and plasticity [120]. It regulates many neurological functions and plays an important role during brain injuries [121]. In 2021, Xiong et al. demonstrated that levels of endogenous BDNF are modulated after HI in a rat model, suggesting it plays an important role in pathogenesis [122]. It has been shown by Cheng et al., in a rat model of HI that a single intracerebroventricular injection of exogenous BDNF (10 µg) just after the insult protected the brain against 50% of tissue loss [123]. In a study in 2003, Galvin et al., showed that the infusion of a low-dose of exogenous BDNF (4.5 µg/day) for 3 days in the striatum significantly increase the survival of medium spiny neuron, a particular neuronal population, by 43% [124]. Unfortunately, in these studies, BDNF was directly delivered into the brain of the animals probably because of its low penetration into the brain through the BBB [123,125]. Validation of the exogenous BDNF as a therapeutic approach in human neonates requires the development of another route of administration.

### 5.4. Neurotrophin-3

Neurotrophin-3 (NTF-3) is a growth factor that play a crucial role in the plasticity and maintenance of the adult nervous system [120,126]. In 2003, Galvin et al., infused a low-dose of exogenous NTF-3 (12 µg/day) continuously for 3 days in the striatum of rats (1 day before surgery and 2 days after). By so doing, the NFT-3 significantly increased the survival of medium spiny neuron, a particular neuronal population, by 33% [124]. There is a need for studies on the efficacy of NTF-3 in protecting/restoring the BBB after hypoxic ischemic events. Moreover, more work is needed on the use of NFT-3 in the treatment of HI-related brain damage.

All these neurotrophins are still under development as are many other molecules and more studies are needed to evaluate the potential effect of exogenous supplementation in the treatment of infants after HI.

## 6. Perspectives

HI is a multifactorial storm in the neonatal period that affects 1 to 8 per 1000 neonates in high-income countries and 1 to 25 per 1000 neonates in low-income countries, accounting for 23% of deaths in the first 4 weeks of life [1]. A huge number of neuroprotective treatments are currently under development to prevent secondary injuries, whereas there is little focus on regenerative therapies. However, besides the reported beneficial use of many drugs in the prevention of neuronal cell death in animal models, supporting data from clinical trials are still lacking. Both inflammation and BBB abnormalities predispose individuals to irreversible neuronal damage. There is an enormous unmet clinical need for effective therapeutic interventions against neonatal HIE. A major limitation of previous unsuccessful approaches is their focus on either the neuronal compartment or the BBB, without accounting for the dynamic interactions between these two biological systems. Several pieces of evidence suggest that changes in cortical blood flow and activity during HI [127] are related to fluctuations in breathing and blood oxygenation, which predominantly affect CNS endothelial cells. Importantly, this suggests that the dysfunction of CNS endothelial cells impairs neuronal activity [128,129]. These facts suggest that the deleterious crosstalk between neurons and CNS endothelial cells of the BBB in HI is associated with worse functional outcomes and should be a central key in drug development.

In this context, there is a real need to target these two compartments through the development of multi-target drugs to improve neuronal plasticity in neonatal HI.

## Figures and Tables

**Figure 1 ijms-25-05449-f001:**
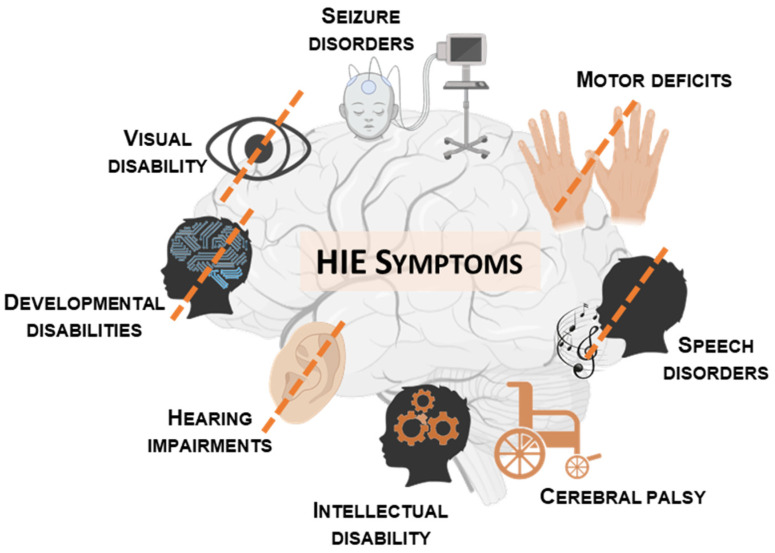
Long-term disabilities developed after hypoxia-ischemia. From the HIE Help Center website.

**Table 1 ijms-25-05449-t001:** Sarnat classification. According to their clinical presentations, patients were classified into three groups: mild, moderate and severe. Physicians use this classification to determine HIE severity groups. From Neonatal Intensive Care Unit Clinic.

Sarnat Grading Scale		Mild	Moderate	Severe
Level of consciousness		Hyperalert	Lethargic	Stuporose
Neuromuscular control	Muscle tone	Normal	Mild hypotonia	Flaccid
	Posture	Mild distal flexion	Strong distal flexion	Intermittent decerebration
	Stretch reflexes	Overactive	Overactive	Decreased or absent
	Segmental myoclonus	Present	Present	Absent
Complex reflexes	Suck	Normal/Weak	Weak/Absent	Absent
	Moro	Strong	Weak/Incomplete	Absent
	Oculovestibular	Normal	Overactive	
	Tonic neck	Slight	Strong	Absent
Autonomic function		Generalized sympathetic	Generalized parasympathetic	Both systems depressed
Pupils		Mydriasis	Miosis	Variable; often unequal; poor light reflex
Heart Rate		Tachycardia	Bradycardia	Variable
Bronchial and Salivary secretions		Sparse	Profuse	Variable
Gastrointestinal motility		Normal or decreased	Increased; diarrhea	Variable
Seizures		Absent	Common	Frequent/difficult to control
Electroencephalogram findings		Normal	Early: Low-voltage continuous delta and theta Later: periodic pattern (awake) Seizures: focal 1-to 1-Hz spike-and-wave	Early: periodic pattern with isopotential phases Later: totally isopotential
Duration		<24 h	2–14 h	Hours to weeks

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
