# Peer review of "Targeting the Multiple Complex Processes of Hypoxia-Ischemia to Achieve Neuroprotection"

_ijms, 2024, doi:10.3390/ijms25105449_

Round 1

Reviewer 1 Report

Comments and Suggestions for Authors

This work reviews and summarizes relevant information regarding the deleterious effects provoked by the lack of oxygenated blood flow in the neonatal period, which gives rise to hypoxic-ischemic encephalopathy (HIE) that strongly affects term and premature newborn infants, and is a major cause of brain damage. Depending on the extent of the HIE lesion a high proportion of asphyxiated infants die early, and those who survive are prone to develop several long-term neurological disabilities. To date, there are very few therapeutic options to treat this affection. The most currently used is hypothermia but it can only be used to treat full term infants, and has several limitations in its application. Thus, the search for alternative pharmacological treatments with other neuroprotective agents (alone or in combination with hypothermia) is necessary.

The authors analyze the main cellular processes and molecular mechanisms that are triggered in response to brain injury during hypoxia-ischemia, that are the basis to attempt the development of several neuroprotective agents. The therapeutic potential of some of these molecules, which have been explored in pre-clinical and clinical trials, are revised in this study.

The information in this work refers to the current and most studied neuroprotective therapies under development to treat HIE. However, it is not extensive. The text could be enriched if it also incorporates other agents that, although still are under study in several animal models, have shown a good potential as neuroprotective factors in response to HI. Among them are growth hormone (GH), insulin-like growth factor-1 (IGF-1), and some local growth factors such as brain-derived neurotrophic factor (BDNF) or neurotrophin-3 (NTF-3), among others, which are also involved in the response to HI. The authors are encouraged to include these agents in their review.

Minor corrections

Several typos were found in the text:

Line 26: it says lowerthan-normal, should say lower than normal or lower-than-normal

Line 28: it says feotal, should say foetal

Line 88: it says we addessed describe the pass and current therapeutic intervention, please revise what you meant (addressed and the past and current therapeutic?)

Lines 103, 268, 290: it says kainite should say kainate

Line 110: it says predictor in the development in of brain injury, should say development of brain injury

Line 201: check the verb conjugation in the context of the sentence: it says “which have”, should say “which has”

Line 208-210: check the paragraph writing, it is not clear. There is a citation to Heyborne et al., instead of a numbered reference and no year is included. It does not appear in the list of references

Line 234: the sentence should initiate as In addition to multiple… not addition to multiple

Line 307: it says discribe should say described

Table 2: it says Allopurinol cross the BBB should say crosses, or as in the other boxes: Able to cross the BBB

Comments on the Quality of English Language

Please correct all the typos mentioned above, and revise the syntaxis in the sentences in Line 88, and in lines 208-210.

Author Response

We thank the reviewer for raising the important point to improve our article.

All the concerns have been addressed and more particularly the use of neurotropic factors for neuroprotection in the context of brain injury.

The paper has been revised carefully and all minor typo have been corrected.

Reviewer 2 Report

Comments and Suggestions for Authors

Comments to the Authors

"Targeting the Various Complex Processes of Hypoxia-Ischemia to Achieve Neuroprotection" was a well-written study.

The work continues to significantly advance the discipline, along with those of other authors.

Below are my thorough observations:

I've discovered through an examination of the literature, especially 2023–2024, that other authors have tackled this subject more than once.

I request an update of the manuscript, referencing the mechanisms of action, among other things, and an update of the literature to make the writers' work more inventive.

The similarity check must be less of 30%.

Author Response

I would like to thank the reviewer for raising important point to improve our paper.

- One of the concern is related to the fact that many reviews have been published on the topic between 2023 and 2024.

- More mechanisms of therapeutic strategies currently undergoing clinical trials have been added in the revised version of manuscript. A chapter on the role of microglia  as well as of the stem cells in neuroprotection during HI has been added. References have been updated

Round 2

Reviewer 1 Report

Comments and Suggestions for Authors

The revised version of this manuscript attended most of the queries raised in the original manuscript and incorporated new information as requested regarding the potential use of  GH, IGF-1, BDNF and NT3 as coadjuvant factors in HIE therapies.

Some typos remain in the manuscript that should be corrected:

-       Line 25, it says “results”, should say “result”

-       Line 28, it says “feotal”, should say “fetal”

-       Line 166, it says “effect oxypurinol”, should say “effect of oxypurinol”

-       Line 352, it says “ivolved”, should say “involved”

-       Line 354, it says “plays a important”, should say “play an important”

-       Line 363, it says “ because oflow pentration”, should say “because of low penetration”